# Circulating platelets modulate oligodendrocyte progenitor cell differentiation during remyelination

Amber R Philp[1,2,3], Carolina R Reyes[1,2,4], Josselyne Mansilla[1,2], Amar Sharma[3], Chao Zhao[3†], Carlos Valenzuela-Krugmann[1,2,4], Khalil S Rawji[3†], Ginez A Gonzalez Martinez[3], Penelope Dimas[3†], Bryan Hinrichsen[1,2], César Ulloa-Leal[1,2,5], Amie K Waller[3,6], Diana M Bessa de Sousa[7], Maite A Castro[2,8], Ludwig Aigner[7], Pamela Ehrenfeld[2,9], Maria Elena Silva[1,2,4], Ilias Kazanis[3,10], Cedric Ghevaert[3,6], Robin JM Franklin[3†], Francisco J Rivera[1,2,4]*

[1]Laboratory of Stem Cells and Neuroregeneration, Institute of Anatomy, Histology and Pathology, Faculty of Medicine, Universidad Austral de Chile, Valdivia, Chile; [2]Center for Interdisciplinary Studies on the Nervous System (CISNe), Universidad Austral de Chile, Valdivia, Chile; [3]Wellcome-MRC Cambridge Stem Cell Institute & Department of Clinical Neurosciences, University of Cambridge, Cambridge, United Kingdom; [4]Translational Regenerative Neurobiology Group (TReN), Molecular and Integrative Biosciences Research Programme (MIBS), Faculty of Biological and Environmental Sciences, University of Helsinki, Helsinki, Finland; [5]Escuela de Ciencias Agrícolas y Veterinarias, Universidad Viña del Mar, Viña del Mar, Chile; [6]Department of Haematology and NHS Blood and Transplant, University of Cambridge, Cambridge, United Kingdom; [7]Institute of Molecular Regenerative Medicine, Paracelsus Medical University, Salzburg, Austria; [8]Instituto de Bioquímica y Microbiología, Facultad de Ciencias, Universidad Austral de Chile, Valdivia, Chile; [9]Laboratory of Cellular Pathology, Institute of Anatomy, Histology & Pathology, Faculty of Medicine, Universidad Austral de Chile, Valdivia, Chile; [10]School of Life Sciences, University of Westminster, London, United Kingdom

*For correspondence:
francisco.rivera@helsinki.fi

Present address: †Altos Labs – Cambridge Institute of Science, Cambridge, United Kingdom

Competing interest: The authors declare that no competing interests exist.

**Abstract** Revealing unknown cues that regulate oligodendrocyte progenitor cell (OPC) function in remyelination is important to optimise the development of regenerative therapies for multiple sclerosis (MS). Platelets are present in chronic non-remyelinated lesions of MS and an increase in circulating platelets has been described in experimental autoimmune encephalomyelitis (EAE) mice, an animal model for MS. However, the contribution of platelets to remyelination remains unexplored. Here we show platelet aggregation in proximity to OPCs in areas of experimental demyelination. Partial depletion of circulating platelets impaired OPC differentiation and remyelination, without altering blood-brain barrier stability and neuroinflammation. Transient exposure to platelets enhanced OPC differentiation in vitro, whereas sustained exposure suppressed this effect. In a mouse model of thrombocytosis (Calr[+/-]), there was a sustained increase in platelet aggregation together with a reduction of newly-generated oligodendrocytes following toxin-induced demyelination. These findings reveal a complex bimodal contribution of platelet to remyelination and provide insights into remyelination failure in MS.

### eLife assessment

This **important** study aims to understand how the regulation of oligodendrocyte progenitor cell (OPC) remyelination and function contributes to the treatment of multiple sclerosis. The authors provide **convincing** evidence for the platelets mediating OPC differentiation and remyelination. This work will be of interest to several disciplines.

## Introduction

In the CNS, remyelination by newly generated oligodendrocytes is largely mediated by the differentiation of oligodendrocyte progenitor cells (OPCs). In response to demyelination, OPCs proliferate, migrate, and differentiate into remyelinating oligodendrocytes (*Franklin and ffrench-Constant, 2008*). Although remyelination represents a robust regenerative response to demyelination, it fails during the progress of multiple sclerosis (MS), a CNS autoimmune demyelinating disease (*Noseworthy et al., 2000*). Unravelling the mechanisms that govern remyelination is essential to our understanding of why this important regenerative process fails in MS, as well as in guiding the development of regenerative therapies.

Platelets are small, anucleate cells essential for haemostatic plug formation (*Semple et al., 2011*). Platelets also display tissue-regenerative properties (*Nurden, 2011*). Several growth factors known to modulate OPCs' responses to demyelination, such as PDGF and FGF2 (*Woodruff et al., 2004*; *Murtie et al., 2005*; *Zhou et al., 2006*; *Clemente et al., 2011*; *Hiratsuka et al., 2019*), are stored in platelets (*Chen et al., 2012*; *Lohmann et al., 2012*; *Schallmoser and Strunk, 2013*; *Warnke et al., 2013*). We have previously shown that platelet lysate increases neural stem / progenitor cells (NSPCs) survival, an alternative but infrequent cellular source for mature oligodendrocytes (*Kazanis et al., 2015*). Although this evidence argues in favour of a beneficial contribution of platelets to remyelination, other studies suggest a detrimental role. CD41-expressing platelets and platelet-contained molecules are found in non-remyelinated MS lesions (*Lock et al., 2002*; *Han et al., 2008*; *Langer et al., 2012*; *Simon, 2012*; *Steinman, 2012*). Moreover, MS patients show increased levels of circulating platelet microparticles (PMPs) (*Marcos-Ramiro et al., 2014*) and the number of PMPs are indicative of the clinical status of the disease (*Sáenz-Cuesta et al., 2014*). Additionally, MS patients display high plasma levels of platelet-specific factors such as, P-selectin and PF4 that correlate with disease course and severity, respectively (*Cananzi et al., 1987*; *Kuenz et al., 2005*). In the animal model for MS, experimental autoimmune encephalomyelitis (EAE), platelet numbers within CNS increase (*D'Souza et al., 2018*). When platelets were immunodepleted before clinical onset, EAE severity is decreased (*Langer et al., 2012*; *Kocovski et al., 2019*). Here, we ask whether circulating platelets regulate OPC function and how this impacts remyelination.

## Results

### Circulating platelets transiently accumulate in response to demyelination and accumulate in close proximity to OPCs

We first assessed the distribution of platelets during remyelination. We created lysolecithin (LPC)-induced demyelinating lesions in the spinal cord white matter of wild type (WT) mice and collected tissue sections at 1-, 3-, 5-, 7-, 10, and 14 days post-lesion (dpl). We observed CD41+ platelet aggregates within and around the lesion early after demyelination (3 dpl) (p-value <0.01; *Figure 1A and B*). However, this was transient as platelet aggregates subsequently decreased until no aggregates were detected at 14 dpl (*Figure 1A and B*). To assess whether platelet recruitment was specific to demyelination we injected PBS containing DAPI directly into the spinal cord. No signs of demyelination were observed under these conditions and platelet aggregation was minimal at 1- and 3 days post-PBS injection (*Figure 1C*). We next evaluated the localization of platelets within the lesion. Large platelet aggregates were found within the blood vessels and within the tissue parenchyma at 5 dpl (*Figure 1D*). Platelets often localized with Olig2+ cells around blood vessels, a scaffold used by OPCs for migration (*Tsai et al., 2016*; *Figure 1D*).

### Depletion of circulating platelets alters OPC differentiation and remyelination in vivo

To investigate whether circulating platelets modulate OPC function in vivo, we used a platelet depletion model (*Figure 2A*). LPC-induced focal demyelinating lesions were performed in WT mice

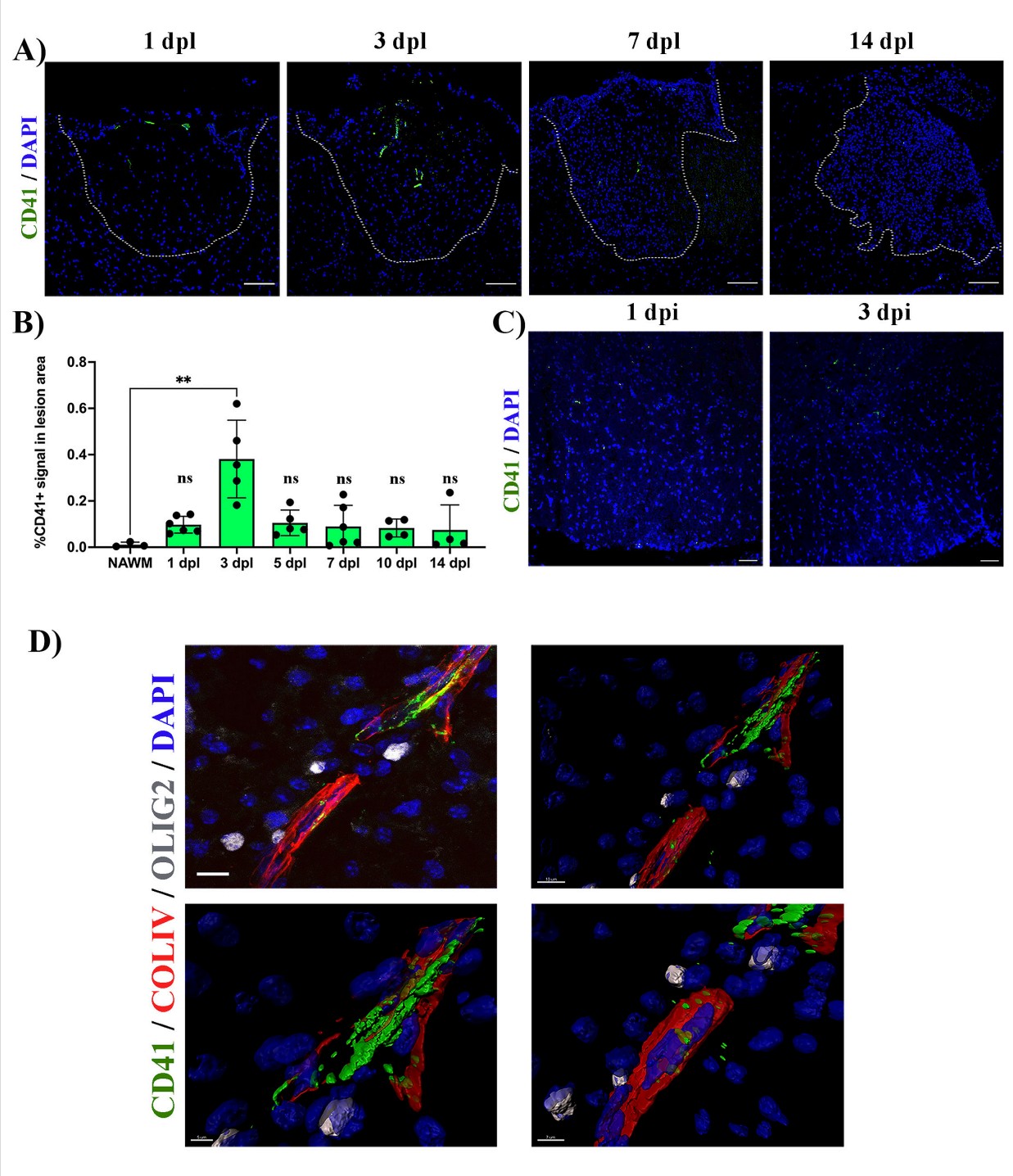

**Figure 1.** Platelets accumulate in response to demyelination. (**A**) LPC induced demyelinating lesions in spinal cord white matter of WT mice at 1, 3, 7, and 14 dpl, stained for platelets (CD41+). Scale bar 100 μm. (**B**) Quantification of CD41+ signal within the demyelinated lesion at 1 (n=6), 3 (n=5), 5 (n=5), 7 (n=6), 10 (n=4), and 14 dpl (n=4), and in NAWM (n=3). (**C**) Platelet staining (CD41+) in spinal cord white matter injected with PBS/DAPI. Scale bar 50 μm. (**D**) Upper left panel: localization of platelets within blood vessels (ColIV+) and in close proximity with OPCs (Olig2+) at 5 dpl. Upper right panel: IMARIS 3D projection shows the spatial distribution of platelets. Scale bar 10 μm. Lower panels: magnification of the IMARIS projection showing platelet aggregation within the blood (left panel) and penetration into the parenchyma (right panel). Scale bars: 5 μm (left panel) and 7 μm (right panel). Data were analysed using a Kruskal Wallis test. Data represent the mean ± SD. ** p<0.01; ns (not significant), p>0.05.

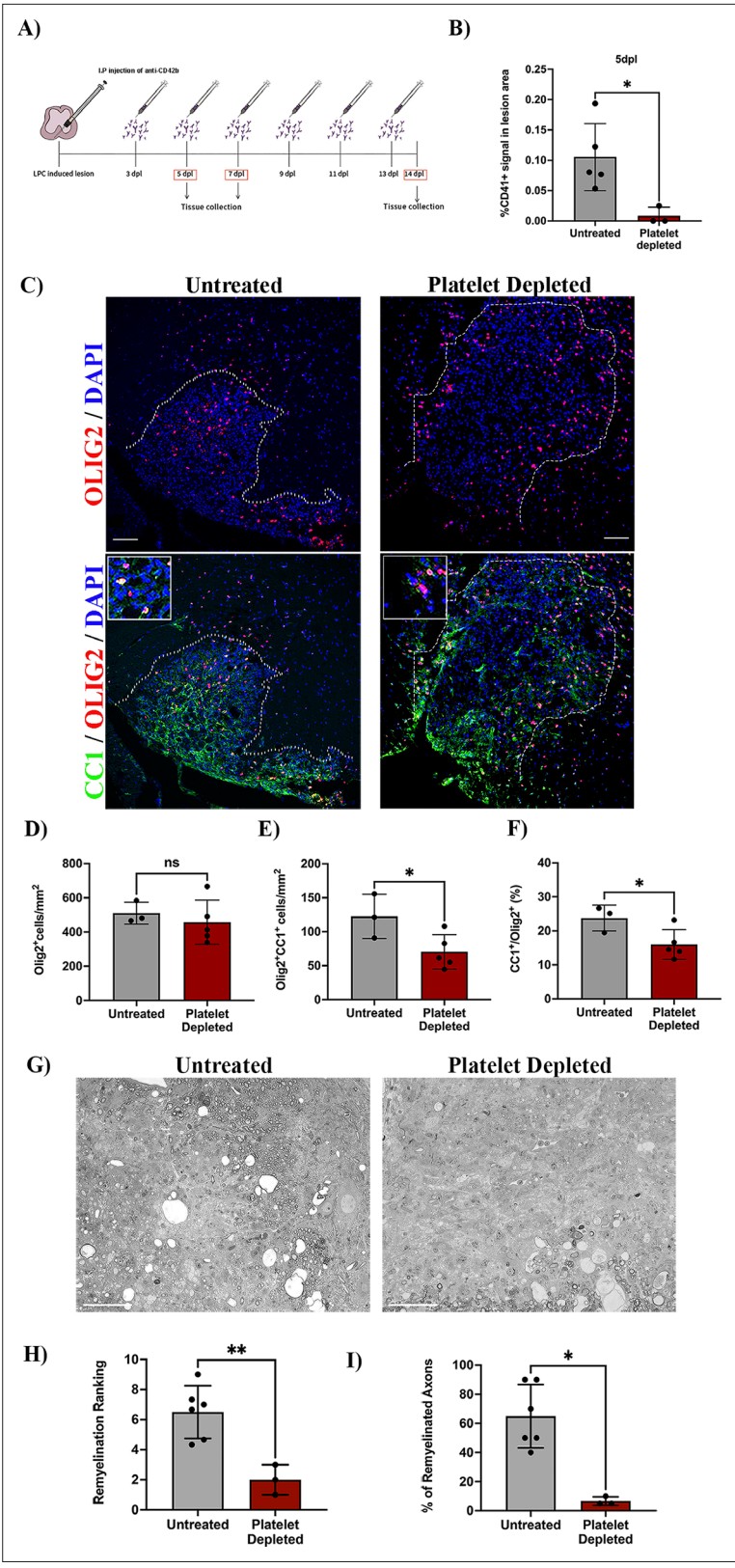

**Figure 2.** Platelet depletion impairs remyelination in vivo. (**A**) Schematic representation of the LPC-induced demyelination model coupled with platelet depletion using anti-CD42b. (**B**) Quantitative analysis of CD41+ signal at 5 dpl in untreated (*n=5*) and platelet depleted mice (*n=3*). (**C**) Representative images of immunofluorescence staining of oligodendroglial lineage cells in untreated and platelet depleted mice at 7 dpl using Olig2+ (upper

*Figure 2 continued on next page*

*Figure 2 continued*

panels) and mature oligodendrocytes using Olig2⁺/CC1⁺ (lower panels). Boxed areas represent high magnification images. (**D–F**) Quantitative analysis of oligodendroglia at 7 dpl in untreated (*n=3*) and platelet depleted mice (*n=5*). (**G**) Representative images of toluidine blue staining of remyelination in untreated (*n=6*) and platelet depleted mice (*n=3*) at 14 dpl and (**H–I**) its quantification by relative ranking analysis. Data were analysed using an Unpaired Student's t-test or Mann-Whitney U test. Data represent mean ± SD. * p<0.05; ** p<0.01; ns (not significant), p>0.05. Scale bars, 100 µm.

The online version of this article includes the following figure supplement(s) for figure 2:

**Figure supplement 1.** Platelet depletion does not alter BBB permeability.

**Figure supplement 2.** Changes in circulating platelet numbers does not alter the macrophage/microglia population during remyelination.

followed by the administration of anti-CD42b at 3 dpl and every second day to prevent further platelet recruitment (*Morodomi et al., 2020*; *de Sousa et al., 2023*). We first confirmed that this depletion strategy leads to decreased numbers of recruited platelets, with no accumulation in the lesion (p-value <0.05; *Figure 2B*). At 7 dpl, there was no difference in the number of Olig2⁺ cells within the lesion between the platelet depleted and untreated group (*Figure 2C*, upper panels, and D), indicating that platelets do not alter OPC recruitment in response to demyelination. Through the detection of CC1 expression, a marker that identifies mature oligodendrocytes (*Figure 2C*, lower panels), we found that platelet depletion significantly decreased the number and percentage of Olig2⁺/CC1⁺ cells compared to untreated mice (p-value <0.05; *Figure 2E and F*), indicating that platelet depletion impairs OPC differentiation. Consistently, at 14 dpl we observed a significant decrease in the extent of remyelination (p-value <0.01; *Figure 2G and H*) and the percentage of remyelinated axons compared to untreated animals (p-value <0.05; *Figure 2I*). Previous studies have shown that decreasing the number of circulating platelets increases blood vessel leakiness (*Cloutier et al., 2012*; *Gupta et al., 2020*). To assess whether impaired OPC differentiation might be due to fibrinogen extravasation (*Petersen et al., 2017*) or enhanced demyelination due to neutrophil infiltration (*Rüther et al., 2017*), we evaluated their presence within the lesion parenchyma after platelet depletion. There were no significant differences between neutrophil (*Figure 2—figure supplement 1A, B*) and fibrinogen extravasation (*Figure 2—figure supplement 1C, D*) after platelet depletion at 7 dpl, indicating that remyelination impairment likely derives from low numbers of circulating platelets rather than increased vascular leakiness.

## Depletion of circulating platelets does not alter macrophage/microglia numbers and polarization during remyelination

Blood-borne macrophages and CNS-resident microglia are essential for OPC differentiation during remyelination (*Kotter et al., 2006*; *Miron et al., 2013*). As platelets regulate macrophage function in neuroinflammation (*Langer and Chavakis, 2013*; *Carestia et al., 2019*; *Rolfes et al., 2020*) and since platelets are located near macrophages/microglia upon demyelination (*Figure 2—figure supplement 2A*), we evaluated whether platelet depletion affects these cell populations (*Figure 2—figure supplement 2B*). At 10 dpl, platelet depletion did not alter the total number of IBA-1⁺ (*Figure 2—figure supplement 2C*), pro-inflammatory IBA-1⁺/CD16/32⁺ (*Figure 2—figure supplement 2D*) or anti-inflammatory IBA-1⁺/Arg-1⁺ (*Figure 2—figure supplement 2E*) macrophages/microglia present within the remyelinating lesion. Furthermore, platelet depletion did not influence macrophage/microglia phagocytic activity as no difference in myelin debris clearance, detected by Oil-Red O, was observed (*Figure 2—figure supplement 2F, G*). Therefore, circulating platelets likely impact OPC differentiation without interfering with macrophage/microglia numbers/polarization during remyelination.

## Transient in vitro exposure to platelets enhances OPC differentiation

To confirm whether transient platelet exposure directly enhances OPC differentiation, OPCs were briefly exposed to washed platelets (WP) for 3 days (pulse) and differentiation was assessed 3 days after WP withdrawal. OPCs briefly exposed to 10% WP exhibited a significant increase in the percentage of Olig2⁺/MBP⁺ mature oligodendrocytes compared to the vehicle treated control (p-value <0.0001; *Figure 3A and B*), indicating that transient contact to platelets directly promotes OPC differentiation.

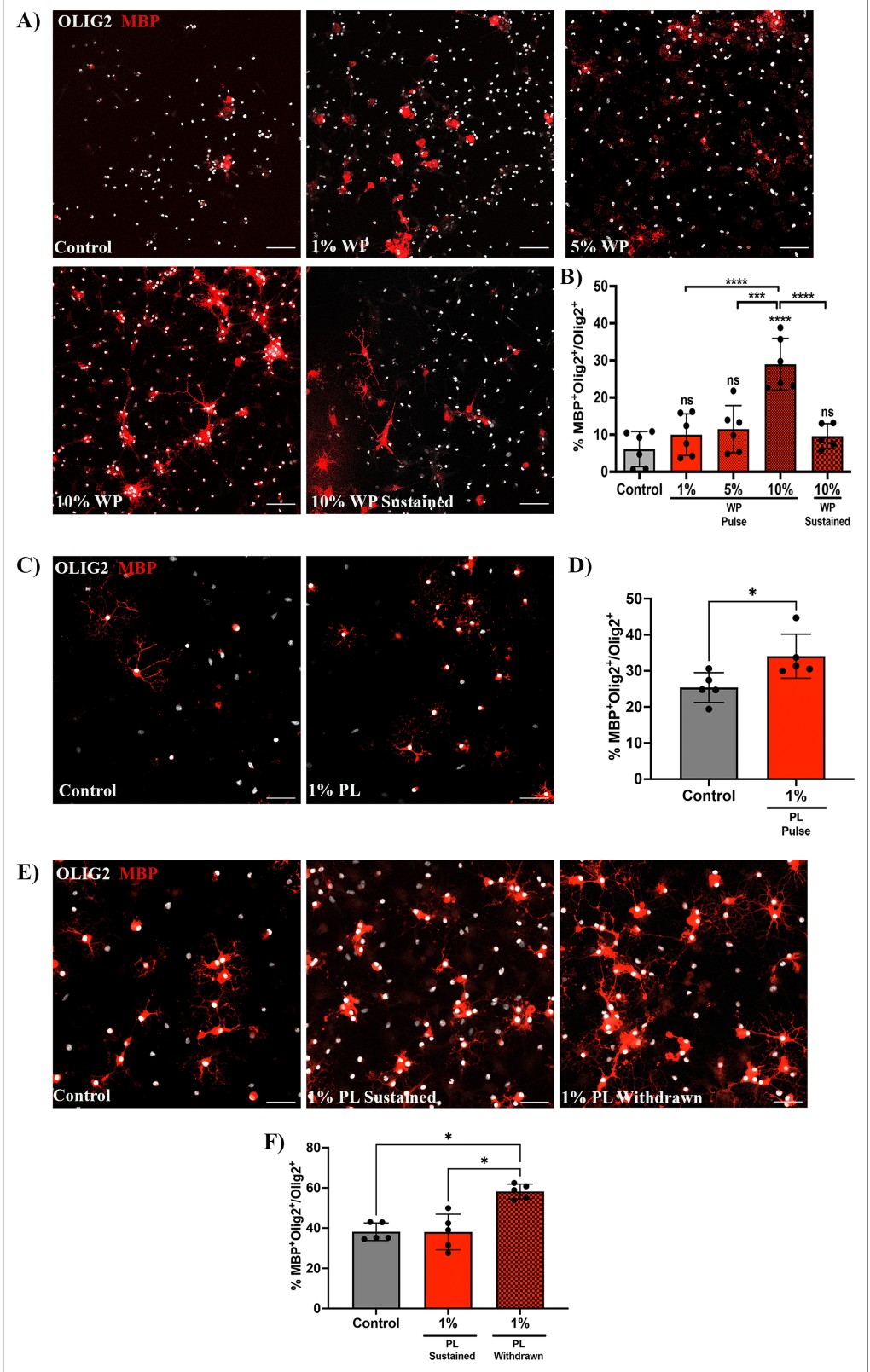

**Figure 3.** Prolonged exposure to platelets suppresses their ability to enhance OPC differentiation.
(**A**) Representative fluorescence images of OPCs co-cultured with 1 (n=6), 5 (n=6), and 10% (n=6) washed platelets (WP) for 3 days in vitro (DIV), followed by WP removal for an additional 3 DIV (Pulse). Additionally, OPCs were co-cultured in the presence of 10% WP for 6 DIV (n=5) (Sustained). Vehicle treated OPCs represents the

*Figure 3 continued*

control condition (n=6). (**B**) Graph represents the percentage of Olig2$^+$MBP$^+$ oligodendrocytes within the total Olig2 population (quantitative analysis of OPC differentiation). (**C**) Representative images of OPCs exposed to 1% platelet lysate (PL) (n=5) for 6 DIV. Vehicle treated OPCs represents the control condition (n=5). (**D**) Graph represents the quantitative analysis of OPC differentiation as in B. (**E**) Representative images of OPCs exposed to either PL for 9 DIV (Sustained) (n=5) or 6 DIV with PL followed by its removal for an additional 3 more DIV (Withdrawn) (n=5). Vehicle-treated OPCs represents the control condition (n=5). (**F**) Graph shows the quantitative analysis of OPC differentiation as in B and D. Data were analysed using one-way ANOVA followed by Tukey's post-hoc test, a Mann-Whitney U test, or Kruskal-Wallis test. Data represent the mean ± SD. * p<0.05; *** p<0.001; **** p<0.0001; ns (not significant), p>0.05. Scale bars, 50 μm.

Similar increases in the proportion of Olig2$^+$/MBP$^+$ mature oligodendrocytes were observed when OPCs were transiently exposed to 1% platelet lysate (PL) compared to vehicle-treated control, indicating that this effect is, at least in part, mediated through platelet-contained factors and direct cell-cell contact is not essential (p-value <0.05; *Figure 3C, D*).

## Sustained increase in circulating platelets hampers OPC differentiation during remyelination

Chronically-demyelinated MS lesions have been reported to contain a substantial number of platelets and their derived molecules (*Lock et al., 2002*; *Han et al., 2008*; *Langer et al., 2012*; *Simon, 2012*; *Steinman, 2012*). To explore the effects of prolonged platelet exposure on OPC differentiation, we conducted experiments with sustained exposure to 10% WP. Contrary to the 3-day pulse-based exposure, 6 days of sustained exposure to 10% WP suppressed the ability of platelets to enhance OPC differentiation (p-value <0.0001; *Figure 3A and B*). Similar findings were observed upon 9 days of sustained exposure to 1% PL (*Figure 3E and F*), indicating effects mediated by platelet-contained factors. To test whether this effect is reversible, PL was withdrawn upon 6 days of sustained exposure, and OPC differentiation was evaluated 3 days later. Interestingly, PL withdrawal rescued the capability of platelets to enhance OPC differentiation when compared to the vehicle-treated control and the sustained condition (p-value <0.05; *Figure 3E, F* - F).

To assess whether a permanent increase of circulating platelets may hamper OPC differentiation during remyelination, we used a conditional mouse knock-in model carrying a mutation within the calreticulin gene in a heterozygous fashion controlled by the Vav1 hematopoietic promoter, resulting in sustained thrombocytosis (2–3 times more circulating platelets) without alterations in other cell lineages (*Li et al., 2018*). We induced a demyelinating lesion by LPC injection in the spinal cord white matter of *Calr*$^{+/-}$ mice and evaluated platelet recruitment and OPC differentiation. As expected, at 5 and 10 dpl, *Calr*$^{+/-}$ mice showed increased levels of circulating platelets (p-value <0.01; *Figure 4B*) as well as a higher number of recruited platelets into the lesion (p-value <0.05; *Figure 4A and C*). At 10 dpl, *Calr*$^{+/-}$ mice displayed a reduced number of mature Olig2$^+$/CC1$^+$ oligodendrocytes (*Figure 4D and F*) and a significant decrease in the percentage of differentiated OPCs (p-value <0.05; *Figure 4G*) compared to WT mice, without alterations in the total number of Olig2$^+$ cells (*Figure 4E*). Additionally, we observed a negative correlation between the number of circulating platelets in *Calr*$^{+/-}$ mice with the number of mature oligodendrocytes (r=–0.87, p-value <0.01) (*Figure 4H*). Similar to the platelet depleted model, effects on OPC differentiation are not mediated by inflammation, as *Calr*$^{+/-}$ mice showed no alterations in macrophage/microglia numbers/polarization during remyelination (*Figure 2—figure supplement 2B–E*). These findings indicate that sustained exposure to platelets directly hampers OPC differentiation during remyelination.

## Discussion

In conclusion, our study reveals that in response to myelin damage platelets transiently accumulate within the vascular niche and locate near OPCs. While transient contact to platelets support OPC differentiation, long lasting exposure to elevated numbers of circulating platelets hampers the generation of oligodendrocytes during remyelination. These findings argue in favour of a beneficial physiological role of platelets in remyelination. However, we also highlight that sustained increased platelet

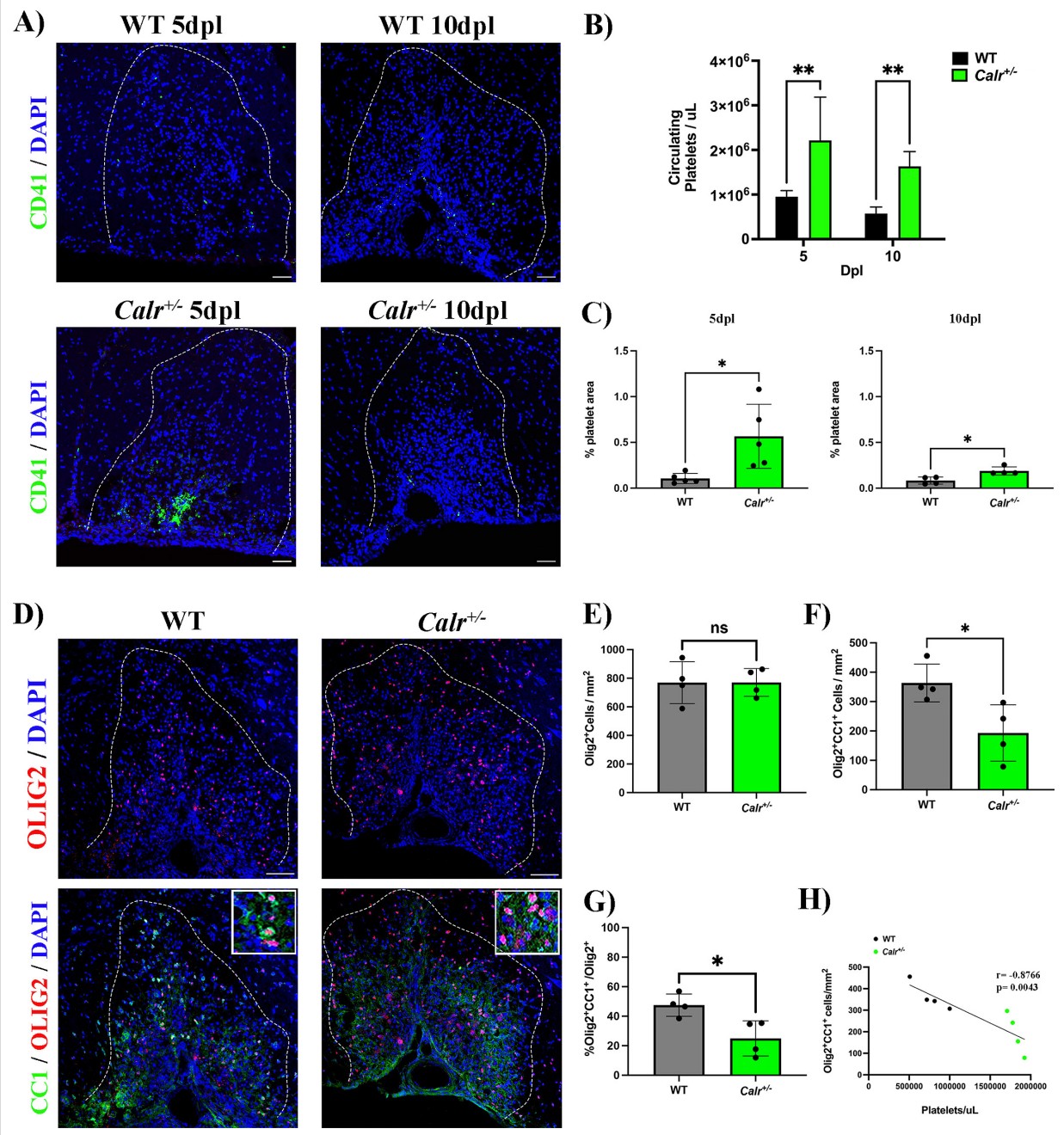

**Figure 4.** A sustained increase in circulating platelets impairs remyelination in-vivo. (**A**) Representative fluorescence images of platelets (CD41+) in LPC induced demyelinating lesions of spinal cord white matter of WT and *Calr* +/-mice at 5 and 10 dpl. Scale bar 50 µm. (**B**) Quantification of circulating platelets in WT vs *Calr* +/-mice at 5 (n=4 and n=5, respectively) and 10 dpl (n=5 and n=6, respectively). (**C**) Quantification of CD41+ signal in demyelinated lesions of WT vs *Calr* +/-mice at 5 dpl (n=5 and n=5, respectively) and 10 dpl (n=4 and n=4, respectively). (**D**) Representative immunofluorescence staining of oligodendroglial lineage cells in untreated and platelet depleted mice at 10 dpl using Olig2+ (upper panels) and mature oligodendrocytes using Olig2+/CC1+ (lower panels) (n=4). Scale bar 100 µm. (**E–G**) Quantitative analysis of oligodendroglia at 10 dpl. (**H**) Correlation between the circulating platelet number with the number of Olig2+/CC1+ cells within the demyelinated lesion. Data were analysed using a two-way ANOVA followed by Bonferroni's post-hoc test, an unpaired t-test, Welch's t-test, a Mann-Whitney U test, or Pearson's correlation coefficient analysis. Data represent the mean ± SD. * p<0.05; ** p<0.01; ns (not significant), p>0.05.

counts, as occurs in MS-related conditions, negatively alter OPC function and contribute to remyelination failure in MS.

Although there is a need to reveal the underlying mechanism(s) by which platelets exert a bimodal action on OPC differentiation, our findings indicate that platelet-contained factors contribute to this effect. This study shows that the regeneration of oligodendrocytes rests on the transient vs sustained presence of platelets within demyelinated lesions. Platelet accumulation in MS lesions may result from blood-brain barrier damage (*Broman, 1964*; *Zlokovic, 2008*) and/or a clearance failure, but changes in their adhesiveness (*Sanders et al., 1968*) and hyperactivity observed during MS (*Sheremata et al., 2008*) may contribute to such scenario. Strategies that restore platelet function, spatially and temporally, represent a future step for developing regenerative therapies in MS.

## Materials and methods

**Key resources table**

| Reagent type (species) or resource | Designation | Source or reference | Identifiers | Additional information |
|---|---|---|---|---|
| Antibody | CD41 rat monoclonal | Abcam | Cat# ab33661; RRID:AB_726487 | Working dilution (1:200) |
| Antibody | CD16/32 rat monoclonal | BD Biosciences | Cat # BD 553142 RRID:AB_394656 | Working dilution (1:200) |
| Antibody | Iba-1 rabbit polyclonal | WAKO | Cat# 019–19741; RRID:AB_839504 | Working dilution (1:500) |
| Antibody | Arg-1 goat polyclonal | Santa Cruz | Cat# sc-18351; RRID:AB_2258542 | Working dilution (1:200) |
| Antibody | NIMP-R14 rat monoclonal | Abcam | Cat# ab2557; RRID:AB_303154 | Working dilution (1:200) |
| Antibody | Olig2 rabbit monoclonal | Abcam | Cat# Ab109186; RRID:AB_10861310 | Working dilution (1:200 in vivo) (1:500 in vitro) |
| Antibody | CC1 mouse monoclonal | Millipore | Cat# OP80; RRID:AB_2057371 | Working dilution (1:1000) |
| Antibody | MBP rat monoclonal | Bio-rad | Cat# MCA409S; RRID:AB_325004 | Working dilution (1:500) |
| Antibody | Collagen IV (ColIV) goat polyclonal | Millipore | CAT# AB769; RRID:AB_92262 | Working dilution (1:100) |
| Antibody | Fibrinogen rabbit polyclonal | Abcam | Cat # ab34269 RRID:AB_732367 | Working dilution (1:200) |
| Chemical compound, drug | L-α-lysophosphatidylcholine | Sigma-Aldrich | Cat # L1381 | Demyelinating agent, Working concentration 1% |
| Chemical compound, drug | CD42b (mixture of rat monoclonal antibodies) | Emfret Analytics; *Evans et al., 2021* | Cat #R300 RRID:AB_2721041 | Platelet depletion antibody, Working concentration 0.6 µg/g |
| Strain, strain background (*Mus musculus*) | Mouse: C7BL/6 | Charles River Laboratories | RRID:SCR_003792 | |
| Strain, strain background (*Mus musculus*) | Mouse: Calr^fl/+:Vav1-Cre mice | *Li et al., 2018* | | |
| Strain, strain background (*Rattus norvegicus*) | Rat: Sprague Dawley | Charles River Laboratories | RRID:SCR_003792 | |

### Animals

All animal work at University of Cambridge complied with the requirements and regulations of the United Kingdom Home Office (Project Licenses PCOCOF291 and P667BD734). All the experiments at Universidad Austral de Chile were conducted in agreement with the Chilean Government's Manual of Bioethics and Biosafety (CONICYT: The Chilean Commission of Scientific and Technological Research, Santiago, Chile) and according to the guidelines established by the Animal Protection Committee

of the Universidad Austral de Chile (UACh). The animal study was reviewed and approved by the *Comité Institucional de Cuidado y Uso de Animales (CICUA)-UACh* (Report Number # 394/2020). All the experiments at University of Helsinki followed the guidelines posed by the Academy of Finland and the University of Helsinki on research ethics and integrity (under Internal License KEK23-022) and accordingly to the National Animal Ethics Committee of Finland (ELLA). Mice and rats had access to food and water ad libitum and were exposed to a 12 hr light cycle. For all in vivo studies animals were grouped randomly (treatment and time) as well as for all in vitro experiments.

## Human subjects

Human platelets were obtained from blood samples of healthy volunteers who signed a consent form before sampling. All procedures were approved by the *Comité Ético y Científico del Servicio de Salud de Valdivia* (CEC-SVS; ORD N° 510) to carry experiments at Universidad Austral de Chile and by the Ethical Committee of the University of Cambridge to perform experiments at this institution. The blood donors at Cambridge were approved by the human biology research ethics committee (reference number: HBREC.2018.13.).

## Focal demyelination lesions

A focal demyelinating lesion was induced in C57BL/6 and *Calr^+/-* mice between 2 and 4 months of age. Animals were anesthetized using Isoflurane/Oxygen (2–2.5%/1000 ml/min $O_2$) and buprenorphine (0.05 mg/kg) was injected subcutaneously immediately before surgery. Local Lysolecithin-driven demyelination in mice was induced as previously described in *Fancy et al., 2009*. Briefly, the spinal cord was exposed between two vertebrae of the thoracic column and demyelination was induced by injecting 1 µL of 1% lysolecithin (L-lysophosphatidylcholine, Sigma) into the ventral funiculus at a rate of approximately 0.5 µl/min⁻¹. The incision was then sutured, and the animal was left to recover in a thermally controlled chamber. Animals were monitored for 72 hr after surgery. Any signs of pain, dragging of limbs, or weight loss of more than 15% of pre-surgery weight, resulted in cessation of the experiment. Mice were sacrificed at 1, 3, 5, 7, 10, and 14 dpl by transcardial perfusion of 4% PFA or glutaraldehyde under terminal anaesthesia.

## Platelet depletion

For platelet depletion, mice received an intraperitoneal injection (IP) of 0.6 µg/g of antiCD42b (Emfret Analytics) (*Evans et al., 2021*), diluted in saline solution, at 3 dpl, followed by IP injections every 48 hr until the end of the experiment period. The effectiveness of platelet depletion was confirmed by measuring the number of circulating platelets using a VetAnalyzer (scil Vet abc Plus). Mice with a circulating platelet number below 200,000 platelets/µL were considered successfully depleted.

## Preparation of washed platelets and platelet lysate

Washed platelets (WP) were prepared as described (*Cazenave et al., 2004*). Briefly, human blood samples were taken from the median cubital vein and collected in sodium citrate followed by centrifugation for 20 min at 120 x *g* to separate the red blood cells from the plasma. Plasma was collected and centrifuged at 1400 x *g* to pellet platelets. Plasma was removed without disrupting the platelet pellet. $PGI_2$ and sodium citrate were carefully added, followed by resuspension in Tyrode's buffer. Platelet number was quantified using a Vet Analyzer and adjusted to a concentration of 1,000,000 platelets/µL. WP were used fresh, meanwhile for the platelet lysate (PL) preparation, the suspension underwent two freeze-thaw overnight cycles. Platelet fragments were then eliminated by centrifugation at 4000 x *g* for 15 min and the supernatant was collected and stored at - 20 °C.

## Primary OPC cultures

OPCs were obtained from Sprague-Dawley postnatal rats (p3 – p6) from both genders. Rat OPCs were isolated and prepared as described by *Neumann et al., 2019*. Cells were then seeded onto glass plates pre-coated with Poly-D-Lysine (PDL) in 24-well plates, with a seeding density of 7000 cells for differentiation assays. For differentiation conditions, T3 was added to the culture media. All experimental conditions were replicated using two independent technical replicates. OPCs were either subjected to various concentrations of washed platelets (1%, 5%, and 10%) or to 1% of platelet lysate of the final volume.

## Histology and immunofluorescence

After transcardial perfusion with 4% PFA, tissue was post-fixed overnight in 4% PFA at 4 °C. After fixation, spinal cords were left in 30% sucrose overnight. Tissue was then embedded in OCT and cut in 15 µm transverse sections on a Leica Cryostat. Samples were stored at - 80 °C until use.

For immunofluorescence staining of tissues, samples were left to thaw for 30 min and washed with PBS. Samples were blocked for 1 hr, using a blocking solution that contained; 10% horse serum, 1% bovine serum albumin, 0.1% cold fish gelatine, 0.1% Triton X-100, and 0.05% Tween 20, diluted in PBS. After blocking, samples were incubated overnight at 4 °C with primary antibody diluted in PBS containing 1% bovine serum albumin, 0.1% cold fish gelatine, and 0.5% Triton X-100. The following primary antibodies were used: rat anti-CD41 (1:200 Abcam), rat anti-CD16/32 (1:200, BD Biosciences), rabbit anti-Fibrinogen (1:200, Abcam), rabbit anti-IBA1 (1:500, WAKO), goat anti-Collagen IV (ColIV) (1:100, Millipore), rabbit anti-Olig2 (1:200, Abcam), goat anti-Arg1 (1:200, Santa Cruz), mouse anti-CC1 (1:1000, Calbiochem), rat anti-NIMP-R14 (1:200, Abcam). Samples were washed three times for 5 min in PBS. After washing, samples were incubated with secondary antibody and DAPI for 1 hr, diluted in the same solution as the primary antibody. Samples were washed three times for 5 min in PBS. Samples were mounted with Fluromount. All secondary antibodies were diluted 1:500. For imaging of spinal cord tissue, the entire lesion area was imaged for five technical replicates.

For immunofluorescence staining of cell cultures, samples were initially washed three times with PBS for 5 min after fixation. The cells were then blocked with 10% Donkey Serum (DKS) in PBS for 1 hr, followed by incubation with the primary antibody overnight, diluted in the same blocking solution. The following primary antibodies were utilized: Rat Anti-Myelin Basic Protein (MBP; 1:500, Bio-Rad) and Rabbit Anti-Oligodendrocyte transcription factor 2 (Olig2; 1:500, Abcam). The cells were then washed three times with PBS 1 x for 5 min, followed by incubation with secondary antibodies, diluted in the blocking solution, for 1 hr. The cells were washed three more times with PBS 1 x for 5 min.

Images were captured using a Leica SP8 Laser Confocal, a Zeiss LSM 980 Confocal or an Olympus IX81FV1000. For cell culture imaging, 8–10 photos per well were quantified for each well using an automated macro in ImageJ/Fiji. For in vivo imaging, three to five photos were quantified per animal by a blinded observer. For tissue image analysis and 3D reconstruction of platelet localisation, ImageJ/Fiji (version 2.1.0/1.53 hr) and Imaris (Bitplane, version 9.3.1, and 9.9.0) were used.

## Oil-Red O staining

To analyse myelin debris clearance, tissue sections were stained with Oil-Red O as previously described by *Kotter et al., 2005*. Briefly, sections were stained with freshly prepared Oil-Red O and incubated at 37 degrees for 30 min. Slides were washed and mounted using an aqueous mounting medium. Image J was used to threshold and quantify Oil-Red O images.

## Remyelination ranking analysis

For remyelination studies, tissue was fixed with 4% glutaraldehyde and embedded in resin. Semi-thin sections of the lesion were cut and stained with Toluidine Blue. Three blinded observers ranked the level of remyelination for each biological individual, giving the most remyelinated individual the highest score, and the individual with the lowest degree of remyelination the lowest. The average for each animal was calculated from the three independent observer rankings.

## Statistical analysis

Statistical analysis was performed using GraphPad Prism 10. In vivo data were obtained from three to six animals per groups (n value). In vitro studies were performed considering, at least, three technical replicates and statistical analysis was performed from five to six independent biological experiments (n value). The distribution of data were first tested using a Shapiro-Wilks test. Two-way ANOVA, One-way ANOVA or a Kruskal Wallis one-way analysis, with the corresponding post-hoc test, were used to compared multiple groups, and a Mann-Whitney U-test, an unpaired t-test or Welch's t-test were used to compare between groups. Pearson's correlation coefficient analysis was used for studies involving data correlation. p-Values were represented as $*<0.05$, $**<0.01$, $***<0.001$, $****<0.0001$.

## Acknowledgements

Special acknowledgments to the *Comité Ético y Científico del Servicio de Salud de Valdivia* (CEC-SVS) for ethical guidance and approval (ORD N° 510). The authors thank the Laboratory of Chronobiology, UACh (led by Dr. Claudia Torres-Farfán) for supporting animal experimentation. We would also like to thank Dr. Alerie Guzman de la Fuente for the Fiji/ImageJ macro for in-vitro quantification. Authors thank the funding support from *Agencia Nacional de Investigación y Desarrollo* (ANID, Chile)-FONDECYT Program Regular Grant Numbers 1201706 and 1161787 (both to FJR), ANID-FONDECYT Program Regular Grant Number 1201635 (to PE), ANID-PCI Program Grant N° REDES170233 (to FJR) and N° REDES180139 (to MAC), ANID-National Doctoral Fellowship N° 21170732 (to ARP), N° 21211727 (to CRR) and N° 21221559 (to CVK). In addition, the authors thank the PROFI 6 N° 336234 of the Research Council of Finland.

## Additional information

### Funding

| Funder | Grant reference number | Author |
| --- | --- | --- |
| Agencia Nacional de Investigación y Desarrollo | Fondecyt Regular Number 1201706 and Number 1161787 | Francisco J Rivera |
| Agencia Nacional de Investigación y Desarrollo | PCI Program Grant Number REDES170233 | Francisco J Rivera |
| Agencia Nacional de Investigación y Desarrollo | PCI Program Grant Number REDES180139 | Maite A Castro |
| Agencia Nacional de Investigación y Desarrollo | National Doctoral Fellowship Number 21170732 | Amber R Philp |
| Agencia Nacional de Investigación y Desarrollo | National Doctoral Fellowship Number 21211727 | Carolina R Reyes |
| Agencia Nacional de Investigación y Desarrollo | National Doctoral Fellowship Number 21221559 | Carlos Valenzuela-Krugmann |
| Agencia Nacional de Investigación y Desarrollo | Fondecyt Regular Number 1201635 | Pamela Ehrenfeld |

The funders had no role in study design, data collection and interpretation, or the decision to submit the work for publication.

### Author contributions

Amber R Philp, Conceptualization, Formal analysis, Funding acquisition, Methodology, Project administration, Validation, Visualization, Writing – original draft, Writing – review and editing, Investigation; Carolina R Reyes, Formal analysis, Funding acquisition, Investigation, Methodology, Validation, Visualization, Writing – review and editing; Josselyne Mansilla, Chao Zhao, César Ulloa-Leal, Methodology, Validation, Writing – original draft; Amar Sharma, Khalil S Rawji, Penelope Dimas, Methodology, Validation; Carlos Valenzuela-Krugmann, Formal analysis, Funding acquisition, Methodology, Project administration, Writing – review and editing; Ginez A Gonzalez Martinez, Methodology; Bryan Hinrichsen, Funding acquisition, Methodology; Amie K Waller, Resources, Validation, Writing – original draft; Diana M Bessa de Sousa, Methodology, Project administration; Maite A Castro, Resources, Supervision, Methodology; Ludwig Aigner, Conceptualization, Resources, Supervision, Methodology; Pamela Ehrenfeld, Resources, Supervision, Methodology, Validation; Maria Elena Silva, Supervision, Methodology, Validation, Writing – original draft; Ilias Kazanis, Conceptualization, Supervision, Methodology, Writing – review and editing; Cedric Ghevaert, Robin JM Franklin, Conceptualization, Resources, Supervision, Methodology, Writing – original draft, Writing – review and editing; Francisco J Rivera,

Conceptualization, Resources, Formal analysis, Supervision, Funding acquisition, Methodology, Project administration, Validation, Visualization, Writing – original draft, Writing – review and editing

### Author ORCIDs
Carolina R Reyes ⓘ https://orcid.org/0000-0002-9005-0313
Carlos Valenzuela-Krugmann ⓘ http://orcid.org/0009-0007-4825-5802
Khalil S Rawji ⓘ https://orcid.org/0000-0002-9687-8330
Maria Elena Silva ⓘ http://orcid.org/0000-0001-6236-7180
Francisco J Rivera ⓘ https://orcid.org/0000-0001-7895-6318

### Ethics

Human platelets were obtained from blood samples of healthy volunteers who signed a consent form before sampling. All procedures were approved by the Comité Ético y Científico del Servicio de Salud de Valdivia (CEC-SVS) (ORD N° 510) to carry experiments at Universidad Austral de Chile and by the Ethical Committee of the University of Cambridge to perform experiments at this institution. The blood donors at Cambridge were approved by the human biology research ethics committee (reference number: HBREC.2018.13.).

All animal work at University of Cambridge complied with the requirements and regulations of the United Kingdom Home Office (Project Licenses PCOCOF291 and P667BD734). All the experiments at Universidad Austral de Chile were conducted in agreement with the Chilean Government's Manual of Bioethics and Biosafety (CONICYT: The Chilean Commission of Scientific and Technological Research, Santiago, Chile) and according to the guidelines established by the Animal Protection Committee of the Universidad Austral de Chile (UACh). The animal study was reviewed and approved by the Comité Institucional de Cuidado y Uso de Animales (CICUA)-UACh (Report Number # 394/2020). All the experiments at University of Helsinki followed the guidelines posed by the Academy of Finland and the University of Helsinki on research ethics and integrity (under Internal License KEK23-022) and accordingly to the National Animal Ethics Committee of Finland (ELLA). Mice and rats had access to food and water ad libitum and were exposed to a 12-hour light cycle. For all in vivo studies animals were grouped randomly (treatment and time) as well as for all in vitro experiments.

Reviewer #1 (Public review): https://doi.org/10.7554/eLife.91757.3.sa1
Reviewer #2 (Public review): https://doi.org/10.7554/eLife.91757.3.sa2
Author response https://doi.org/10.7554/eLife.91757.3.sa3

---

## Additional files

### Supplementary files
• MDAR checklist

### Data availability
All source data and pipelines for in vitro analysis can be found at: (1) https://osf.io/8c9ef/?view_only=253f4298cf1440879f772329662419c1 and (2) https://etsin.fairdata.fi/dataset/682e8b1f-2de3-4f8b-87fa-a3d157eda6c1.

The following datasets were generated:

| Author(s) | Year | Dataset title | Dataset URL | Database and Identifier |
|---|---|---|---|---|
| Philp AR, Reyes CR, Mansilla J, Sharma A, Zhao C, Valenzuela-Krugmann C, Rawji KS, Gonzalez Martinez GA, Dimas P, Hinrichsen B, Ulloa-Leal C, Waller AK, Bessa de Sousa DM, Castro MA, Aigner L, Ehrenfeld P, Silva ME, Kazanis I, Ghevaert C, Franklin RJM, Rivera FJ | 2024 | Circulating Platelets Modulate Oligodendrocyte Progenitor Cell Differentiation During Remyelination | https://doi.org/10.17605/OSF.IO/8C9EF | Open Science Framework, 10.17605/OSF.IO/8C9EF |
| Philp AR, Reyes CR, Mansilla J, Sharma A, Zhao C, Valenzuela-Krugmann C, Rawji KS, Gonzalez Martinez GA, Dimas P, Hinrichsen B, Ulloa-Leal C, Waller AK, Bessa de Sousa DM, Castro MA, Aigner L, Ehrenfeld P, Silva ME, Kazanis I, Ghevaert C, Franklin RJM, Rivera FJ | 2024 | Circulating Platelets Modulate Oligodendrocyte Progenitor Cell Differentiation During Remyelination | https://doi.org/10.23729/47130a30-4d3c-44ba-90fd-401ba4909213 | Fairdata IDA Etsin, 10.23729/47130a30-4d3c-44ba-90fd-401ba4909213 |

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
