## [Editor Report · eLife assessment]

This **important** study aims to understand how the regulation of oligodendrocyte progenitor cell (OPC) remyelination and function contributes to the treatment of multiple sclerosis. The authors provide **convincing** evidence for the platelets mediating OPC differentiation and remyelination. This work will be of interest to several disciplines.

---

## [Referee Report · Reviewer #1 (Public review)]

Summary:

The authors have studied the effects of platelets in OPC biology and remyelination. For this, they used mutant mice with lower levels of platelets as a demyelinating/remyelinating scenario, as well as in a model with large numbers of circulating platelets.

Strengths:

-The work is very focused, with defined objectives.

-The work is properly done.

Revision comments:

Having consulted the new version of the work by Amber et al., the modifications and the point-by-point cover letter explaining them give direct answers to my previous comments.

---

## [Referee Report · Reviewer #2 (Public review)]

Summary:

This paper examined whether circulating platelets regulate oligodendrocyte progenitor cell (OPC) differentiation for the link with multiple sclerosis (MS). They identified that the interaction with platelets enhances OPC differentiation although persistent contact inhibits the process in the long-term. The mouse model with increased platelet levels in the blood reduced mature oligodendrocytes, while how platelets might regulate OPC differentiation is not clear yet.

Strengths:

The use of both partial platelet depletion and thrombocytosis mouse models gives in vivo evidence. The presentation of platelet accumulation in a time-course manner is rigorous. The in vitro co-culture model tested the role of platelets in OPC differentiation, which was supportive of in vivo observations.

Revision comments:

Although the mechanisms are limited, the authors addressed the major experiments I suggested.

---

## [Author Response]

The following is the authors’ response to the original reviews.

**Reviewer #1 (Public Review):**
Summary:The authors have studied the effects of platelets in OPC biology and remyelination. For this, they used mutant mice with lower levels of platelets as a demyelinating/remyelinating scenario, as well as in a model with large numbers of circulating platelets.Strengths:-The work is very focused, with defined objectives.-The work is properly done.Weaknesses:-There is no clear effect on a single cell type and/or mechanism involved.

We appreciate the reviewer’s feedback. We understand that from our in vivo studies we are unable to distinguish whether the effects of platelets are directly exerted on OPCs or indirectly through a different cell type. However, data obtained from the platelet depleted model as well as the new data provided in this revised version in CALRHet mice indicate that, at least, macrophages / microglia do not contribute to the observed effects in OPCs. In addition to this, in vitro data support the direct effects of platelets on OPC function.

**Reviewer #2 (Public Review):**
Summary:This paper examined whether circulating platelets regulate oligodendrocyte progenitor cell (OPC) differentiation for the link with multiple sclerosis (MS). They identified that the interaction with platelets enhances OPC differentiation although persistent contact inhibits the process in the longterm. The mouse model with increased platelet levels in the blood reduced mature oligodendrocytes, while how platelets might regulate OPC differentiation is not clear yet.Strengths:The use of both partial platelet depletion and thrombocytosis mouse models gives in vivo evidence. The presentation of platelet accumulation in a time-course manner is rigorous. The in vitro co-culture model tested the role of platelets in OPC differentiation, which was supportive of in vivo observations.Weaknesses:How platelets regulate OPC differentiation is not clear. What the significance of platelets is in MS progression is not clear.

We thank reviewer’s view and assessment of our manuscript. We understand both of the reviewer’s concerns. Firstly, we performed additional in vitro studies and we have confirmed that platelet-contained factors are, at least in part, responsible for modulating OPC differentiation and, thus, direct cell-cell contact is not essential. Secondly, in this revised version, we added references arguing that the plasma levels of platelet microparticles and platelet-specific factors correlate with MS progression and severity.

**Reviewer #1 (Recommendations For the Authors):**
To ameliorate the quality of their work and make it suitable for its publication in eLife, I strongly suggest the authors to:(1) In vitro co-culture platelets and OPCs to check the effects on this latter cell type biology.Response: We have performed in vitro studies, in which OPCs were co-cultured with washed platelets (WP). We observed that OPC differentiation was boosted after a short exposure to WP, however, prolonged exposure to WP suppressed this effect (revised Figure 3A and B). Also, our new data using platelet lysate (PL) indicate that platelet-contained molecules are responsible for this effect (revised Figure 3C and D). Finally, we showed that by removing PL after sustained exposure (6 DIV) the ability of platelets to promote OPC differentiation is rescued (revised Figure 3E and F).(2) In the CALR model, can the authors check effect of chronic exposure to large numbers of platelets? Is this affecting macrophages (including their polarization)?Response: Yes, compared to wild type mouse, in the CALRHET model we confirmed the presence of larger number of platelets within demyelinated lesions (Figure 4A and C). Also, in this revised version we added data showing in the CALRHET model that thrombocytosis does not affect macrophage / microglia numbers and polarization (revised Supplementary Figure 2).(3) Some aspects of the Introduction section seems too old-fashioned (ex.: the use of bFGF instead of FGF2 to refer to Fibroblast Growth Factor 2), as well as it would be convenient to include more recent references on the role of FGF2 and PDGFa in OPC biology.Response: We agree with the reviewer. In this revised version we have changed bFGF for FGF2 and we added more recent references addressing the role of FGF2 and PDGFa in OPC biology.(4) There are some constructions and typos that could be corrected.Response: We have checked language constructions as well as typos, and these have been corrected.(5) Please revise spelling of some names in the bibliography list (ex.: the correct surname is ffrenchConstant, not Ffrench-Constant).

We have revised the spelling of names within the bibliography, and we have corrected them accordingly.

**Reviewer #2 (Recommendations For the Authors):**
Mechanisms of platelet-OPC interactions- transwell co-culture assay will examine if the OPC phenotype is through direct or indirect interactions with platelets;

We have performed additional in vitro studies, in which OPCs were exposed to platelet lysate (PL). New results indicate that a short exposure to PL can promote OPC differentiation (revised Figure 3C and D), while a sustained exposure supresses this effect (revised Figure 3E and F). These findings indicate that platelet-contained factors are, at least in part, responsible for modulating OPC differentiation and, thus, direct cell-cell contact is not essential for such an effect.

- can you revert the phenotype of OPCs co-cultured long with platelets (maturation blocked) by removing platelet (then OPC differentiate again?) to see if the phenotype is reversible or not?

We would like to thank the reviewer for bringing up this interesting question. We have performed additional in vitro studies to address this issue. We found that by removing PL upon 6-days of sustained exposure rescues the ability of platelets to promote OPC differentiation (revised Figure 3E and F). These findings indicate that the supressing effect of prolonged exposure to platelets in OPC differentiation is reversible.

Clinical correlation- How many MS patients has an abnormal number of or exposure to platelets?We have added new information in the introduction section. Indeed, previous studies have shown that MS patients display higher levels of circulating platelet microparticles (PMPs) (MarcosRamiro et al., 2014) as well as increased plasma levels of platelet-specific factors such as, P-selectin and PF4 (Cananzi et al., 1987; Kuenz et al., 2005).do platelets amount correlate with diseases severeness?

We have added new information in the introduction section. Indeed, PMPs are indicative of the clinical status of the disease (Saenz-Cuesta et al., 2014). Also, plasma levels of P-selectin and PF4 correlate with disease course and severity, respectively (Cananzi et al., 1987; Kuenz et al., 2005).

Image quantification- please state how many sections were counted. How many animals were used per condition. Is the practice of blinded observers done for each dataset?

We added this information in the figure legends and in methods section. We counted between 3-5 sections per lesion. We used 3 to 6 animals per experimental group and data was analysed by blinded observers.